# Chemical Composition and Allelopathic Effect of Essential Oil of *Litsea pungens*

Qingbo Kong, Lijun Zhou , Xiaoju Wang, Siyuan Luo, Jiajia Li, Hanyong Xiao, Xinyao Zhang, Tingting Xiang, Shiling Feng, Tao Chen, Ming Yuan and Chunbang Ding *

College of Life Science, Sichuan Agricultural University, Ya'an 625014, China; KQB666666@163.com (Q.K.); zhoulijun@sicau.edu.cn (L.Z.); wxj99537@163.com (X.W.); luosiyuan1998@163.com (S.L.); Ljiajia1118@163.com (J.L.); Xiaohanyongtcm@163.com (H.X.); zxysicau@163.com (X.Z.); Xiangtingting01@163.com (T.X.); fengshilin@outlook.com (S.F.); chentao293@163.com (T.C.); yuanming@sicau.edu.cn (M.Y.)
* Correspondence: dcb@sicau.edu.cn

**Abstract:** Natural plant resources with herbicidal activity may be substitutes for synthetic chemical pesticides, likewise aromatic plant extracts, especially essential oils. Essential oil from *Litsea pungens* has been proved to possess a strong antibacterial property. Interestingly, we found the essential oil also showed a strong allelopathic capacity. Therefore, in the present work, the chemical composition of the essential oil from the fruit of *L. pungens* was analyzed by gas chromatography–mass spectrometry (GC–MS). The weed control abilities of the essential oil were also further evaluated. The results show that the yield of essential oil extracted by steam distillation was 1.4%, and 17 compounds, mainly terpenoids, were identified by GC–MS. In allelopathic tests, the essential oil exhibited a negative effect on seed germination rate and seedling growth of *Lolium perenne* and *Bidens pilosa*. Moreover, chlorophyll content, malondialdehyde content, electrolyte leakage, catalase, superoxide dismutase and peroxidase of seedlings treated with essential oil were also negatively affected. This work could provide a better understanding of the rational utilization of *L. pungens* essential oil for crop cultivation and further development of environment-friendly herbicides.

**Keywords:** *Litsea pungens*; essential oil; allelopathic; bioherbicides

## 1. Introduction

In recent years, the excessive abuse of chemical synthetic herbicides has had a negative impact on environmental pollution and human health, and has led to increasing herbicidal resistance in many weeds [1–3]. In agriculture, weeds constantly compete with crops for water, light and nutrients, which is one of the main causes of crop yield decline. Allelopathic weed control is one of the alternative strategies for weed management in agro-ecosystems, which can reduce the use of traditional herbicides [4]. Compared with traditional herbicides, aromatic plant essential oils have more ecological advantages in the environment and can be used as biological herbicides [5–7]. It is found that essential oils such as clove oil and *eucalyptus* oil are a promising natural resource that can be used as insect repellents, herbicides, food protectors and for environmental protection as non-toxic chemical pesticide alternatives [8–10]. Aghbash reported that essential oils contain bioactive compounds, such as monoterpenes and sesquiterpenoids [11], which belong to the allelochemicals, and are involved in many metabolic and ecological processes [12]. Concomitantly, many reports are interested in the development and utilization of plant essential oils as environment-friendly herbicides [5,13,14].

*Litsea pungens* is a species belonging to the *Litsea* genus and family *Lauraceae*, predominantly distributed in Hubei, Hunan, Guangxi, Sichuan, Guizhou, Yunnan in China. It is a pioneer herb traditionally utilized in medicine [15]. It is reported that *L. pungens* essential oil is a safe and environment-friendly raw material with good bactericidal activity.

*L. pungens* fruit contains essential oil, which can be used as food flavor and cosmetics flavor, and has been widely used as raw material of advanced spices, Vitamin A, as well as violet ketone in industry [16]. So far, many scholars have conducted extensive research on the composition of this *L. pungens* essential oil, mainly focusing on antibacterial, antioxidant, insect inhibition, and cancer inhibition [17–19], while research on inhibiting the growth of weeds and the development and utilization of new herbicidal active substances are very limited.

Therefore, growth parameters as well as physiological and biochemical indicators of weed seeds treated with *L. pungens* essential oil by petri dishes and pot experiments were measured. To the best of our knowledge, this is the first systematic report about the potential allelopathic effect of essential oil from *L. pungens*.

## 2. Materials and Methods

### 2.1. Plant Materials

The fruits of *L. pungens* were gathered from Hongya county (29°56′75.28″ N; 103°24′33.55″ E), Sichuan, in 2019. Fresh fruit dries naturally to get dried fruit. Seeds of *Lolium perenne* (*L. perenne*) and *Bidens pilosa* (*B. pilosa*) were purchased from local seed stores.

Ethanol, Tween 80, ferrous sulfate and sodium sulfate were purchased from Chengdu Kelong Chemical Factory (Chengdu, China). All chemicals were analytical grade and used without further purification.

### 2.2. Isolation of the Essential Oil

The dried fruits of *L. pungens* were ground into fine powder and extracted by steam distillation for 6 h, respectively. After standing and layering, anhydrous sodium sulfate was added to remove water, then the essential oil was obtained and stored in a brown bottle at 4 °C until analyzed by gas chromatography–mass spectrometry and bioassays. According to the following formula to calculate the extraction yield of essential oil:

Oil yield (%, *v/w*) = volume of essential oil (mL)/dried weight of plant material (g) × 100%

### 2.3. Gas Chromatography–Mass Spectrometry Analysis

The volatile oil was analyzed qualitatively and semi-quantitatively by GC–MS. Typically, 1 μL essential oil was resolved in chromatography n-hexane (ratio of 1/10) and n-hexane was used as blank control. Gas chromatographic conditions: HP-5MS column (30 cm × 0.25 mm × 0.25 μm), using a temperature program. This was held at 60 °C for 5 min and then raised to 100 °C at the speed of 12 °C/min keeping 5 min. The temperature was raised to 160 °C at the speed of 10 °C/min and kept for 5 min. Subsequently, the temperature was increased to 280 °C at the speed of 8 °C/min and kept for 5 min. Helium was invoked as the carrier gas and the injection volume was 1 μL with no shunt. The injection port temperature was 260 °C, and the interface temperature was 220 °C. Mass spectrometry conditions: electron bombardment (EI) ion source. Electron energy was 70 eV and the electron multiplier voltage was 1.5 kV. Mass scanning ranged from 40 to 650 (m/z) full scanning [20]. In order to make the experimental spectrum closer to the library spectrum, the standard spectral modulation was used.

The essential oil qualitative identification was based on a comparison of mass spectrometry with standards recorded by the National Institute of Standards and Technology (NIST) libraries. The properties of the compounds were verified by mass spectrometry and linear retention index compared with literature data [21]. The relative proportion of compounds was determined by area standardization method and expressed by area %.

### 2.4. Allelopathic Activity Assays

#### 2.4.1. Seed Germination Experiment (Petri Dish Experiment)

The allelopathic effect of essential oil was determined by a Petri dish method with slight modifications [22]. Briefly, essential oil was decomposed in Tween 80 with different

concentrations. Then essential oil solution was added into a petri dish to dissolve essential oil with different concentrations (0.125–2.000) mg/mL in Tween 80 (The final concentration was 1%). The seeds were disinfected by NaClO (0.5%) for 10 min and rinsed multiple times with distilled water. Distilled water and 1% Tween 80 were used as blank control and negative control, respectively. Two pieces of sterilized filter papers were placed in a sterilized petri dish with a diameter of 9 cm. 5 mL essential oil solution was added in the test group, and 5 mL distilled water and 5 mL 1% Tween 80 solution were added in the control group. After the filter paper was soaked, 30 seeds of *L. perenne* and *B. pilosa* were evenly spread on the filter paper, then the petri dish was covered immediately, and placed in an artificial climate chamber of $(25 \pm 2)$ °C for culture. To ensure the wetness of the filter paper, 2 mL of the test group solution and the control group solution were added to the Petri dish every 2 days after 3 days of culturing. The standard for seed germination is that the radicle or hypocotyl can break through the seed coat by 1–2 mm. The number of plant seeds germinating was observed and recorded every day for 7 consecutive days (until no more germination for 3 consecutive days). The shoot length and root length were measured with a digital display Vernier caliper on the 7th day. Each treatment was repeated 6 times.

The final germination rate, germination force, and germination index were calculated according to the following formula:

$$\text{Germination rate} = (A/A0) \times 100\%$$

where, A = the total number of germinated seeds, A0 = the total number of tested seeds.

$$\text{Germination force} = B/A0$$

where, B = the number of seeds germinated at the peak of germination, A0 = the total number of tested seeds.

$$\text{Germination index} = \sum(Gt/Dt)$$

where, Gt = the number of germinations on the t day, Dt = the number of corresponding germination days.

### 2.4.2. Pot Experiment

The further allelopathic effect of essential oil was assessed with pot assay according to the previous experiment with slight modifications [23]. In a small pot (height 7.5 cm and width 6.0 cm), 185 g of sterilized weed-free seed soil was filled, and 30 sterilized seeds were evenly sown in a pot; 10 mL of essential oil and control solutions were applied to water the pots for 2 weeks, respectively. After the emergence of the seedlings, a sprayer was used to spray 5 mL of essential oil and control solutions on each pot for one week, respectively. This was repeated for each treatment group six times. After spraying for a week, plant-related growth parameters and biochemical indicators were determined.

### 2.4.3. Determination of Photosynthetic Pigments

The content of photosynthetic pigments was determined according to the previous method with slight modifications [24]. We placed 0.10 g leaves into a mortar, and 2 mL of 95% ethanol was added to grind thoroughly with some appropriate amount of quartz sand and calcium carbonate. The mixture was then transferred to a 2 mL centrifuge tube and centrifuged at the speed of 10,000 rpm for 1 min. The supernatant was mixed with 95% ethanol to 50 mL and then the absorbance of the solution was determined at three wavelengths of 663 nm, 645 nm, and 440 nm, respectively:

$$\text{The content of chlorophyll a (mg/g FW)} = (12.72A_{663} - 2.59A_{645}) \times V/Fw \times 1000$$

$$\text{The content of chlorophyll b (mg/g FW)} = (22.88A_{645} - 4.67A_{663}) \times V/Fw \times 1000$$

$$\text{The content of total chlorophyll (mg/g FW)} = (Ca + Cb)$$

The content of carotenoids (mg/g FW) = $(4.7A_{440} - 0.27(Ca + Cb)) \times V/Fw \times 1000$

where, V = the extract volume, Fw = the sample quality.

### 2.4.4. Determination of Antioxidant Enzyme Activity

The activities of enzymes were determined according to the previous method with some modifications [25]. 0.1 g of potted leaves were homogenized in 2 mL phosphate buffer (50 mM and pH = 7.8) and some quartz sand. The mixture was centrifuged at 12,000 rpm at 4 °C for 15 min, and the supernatant was obtained for determining the activity of CAT, POD, and SOD.

We mixed 1.95 mL of phosphate buffer (50 mM and pH = 7), 1 mL of $H_2O_2$ (0.2%), and 50 μL of supernatant for 40 s and the absorbance of the reaction solution was immediately determined at 240 nm. The activity of CAT was determined according to the following formula:

$$\text{Activity of CAT (U/g Fw)} = (A_{240} \times V_t)/(W \times V_s \times 0.01 \times t)$$

We added 2 mL of $H_2O_2$ (0.2%) and 0.95 mL of guaiacol to 10 μL of supernatant solution for 40 s. Then the absorbance of the reaction solution was determined at 470 nm to calculate the activity of POD.

$$\text{Activity of POD (U/g Fw)} = (A_{470} \times V_t)/(W \times V_s \times 0.01 \times t)$$

We mixed 150 μL of riboflavin (0.3 μmol/L) with 2.5 mL of methionine (13 mmol/L) and 250 μL of nitronitroblue tetrazolium chloride. Then 50 μL supernatant was added and the solution was placed in the light incubator for 20 min reacting under 4000 lux light. The non-illuminated control tube contained only buffer solution, placed in the dark as the zero adjustment. The absorbance of each reaction solution was determined at 560 nm to evaluate the activity of SOD.

$$\text{Activity of SOD (U/g Fw)} = ((A_{CK} - A_E) \times V)/(1/2A_{CK} \times W \times V_t)$$

### 2.4.5. Determination of the Content of Malondialdehyde (MDA) and Electrolyte Leakage (EL)

We ground 0.1 g leaves with 5 mL 5% triclosan abrasion in the presence of quartz sand. Then the extract solution was centrifuged at the speed of 3000 rpm for 10 min. 2 mL supernatant was mixed with 2 mL thiobarbituric acid (0.67%, $w/v$), then placed at 100 °C for 30 min. The absorbance of the reaction solution was measured at the wavelengths of 600 nm, 532 nm, and 450 nm, respectively.

$$\text{Content of MDA (mmol/g Fw)} = (6.452 \times (A_{532} - A_{600}) - 0.559 \times A_{450}) \times V_t/V_s \times W$$

We placed 0.1 g leaves in a graduated test tube of 10 mL distilled water, covered with a glass stopper and soaked in a dark room temperature for 24 h. The conductivity (R1) was firstly measured with a conductivity meter, then the tube was heated in a boiling water bath for 30 min. After cooling to room temperature and shaking, the conductivity (R2) was determined again. The EL was calculated according to the following formula:

$$EL = (R1/R2) \times 100\%$$

### 2.5. Statistical Analysis

All the experimental groups were conducted in triplicates. The outcomes were expressed as mean ± standard deviation (SD). Graphpad Prism 8.0 was used for statistical analysis and making graphic pictures. The significance test was analyzed by SPSS 19.0 using the analysis of variance (ANOVA) method and the significance level was considered as $p < 0.05$.

## 3. Results

### 3.1. Chemical Characterization Analysis of L. Pungens Essential Oil

The yield of *L. pungens* essential oil extracted by steam distillation was 1.4% (*v/w*), the density was 0.9 g/mL, which was based on dried weight, and the constituents of *L. pungens* essential oil were analyzed by gas chromatography-mass spectrometry (GC–MS). As shown in Table 1, 17 compounds were detected in the *L. pungens* essential oil, equivalent to 99.54% of the essential oil, Monoterpenes (90.43%) dominated the essential oil, and citral (29.44%) had the highest proportion of monoterpenes. Moreover, sesquiterpenes accounted for 4.68%, which were β-caryophyllene (3.74%), humulene (0.34%) and caryophyllene oxide (0.60%).

**Table 1.** The chemical composition of the *L. pungens* essential oil.

| NO. | Retention Time (min) | Area (%) | Identified Compounds | Type of Compounds | Retention Indices |
|---|---|---|---|---|---|
| 1 | 6.41 | 4.32 | α-Pinene | DM | 922 |
| 2 | 6.76 | 1.50 | Camphene | DM | 943 |
| 3 | 7.37 | 1.10 | Bicyclo(3,1,0)hexane,4-methylene-1-(1-methylethyl)- | DM | 964 |
| 4 | 7.44 | 3.23 | β-Pinene | DM | 967 |
| 5 | 7.69 | 4.43 | 6-methyl-5-Hepten-2-one | Ketone | 958 |
| 6 | 7.80 | 3.73 | Myrcence | CM | 979 |
| 7 | 8.75 | 30.29 | Limonene | MM | 1018 |
| 8 | 8.81 | 5.93 | Eucalyptol | DLM | 1023 |
| 9 | 10.52 | 6.38 | Linalool | CM | 1081 |
| 10 | 11.83 | 1.09 | Citronellal | CM | 1132 |
| 11 | 12.44 | 0.37 | 4-methyl-1-(1-methylethyl)-3-Cyclohexen-1-ol | MA | 1160 |
| 12 | 12.55 | 1.41 | Cis-verbenol | DLM | 1127 |
| 13 | 12.76 | 1.64 | α-Terpineol | MA | 1172 |
| 14 | 14.73 | 29.44 | neral | Cm | 1241 |
| 15 | 18.57 | 3.74 | β-caryophyllene | DS | 1424 |
| 16 | 19.23 | 0.34 | Humulene | SS | 1456 |
| 17 | 21.77 | 0.60 | Caryophyllene Oxide | BS | 1576 |

DM: Dicyclic monoterpenes; MM: Monocyclic monoterpenes; CM: Chain monoterpenes; DLM: Double loop monoterpenes; Cm: Chain monoterpenes; MA: Monoterpene alcohol; DS: Dicyclic sesquiterpenes; SS: Single sesquiterpenes; BS: Bicyclic sesquiterpenoids.

### 3.2. Effect of L. Pungens Essential Oil on Germination and Seedling Growth (Petri Dish Experiment)

The allelopathic effect of *L. pungens* essential oil on seed germination of two plants (*L. perenne* and *B. pilosa)* was evaluated (Figure 1A–C). The germination rate and germination force of seeds are one of the main index reflecting the seed quality. As shown in Figure 1A,B, the essential oil inhibited ($p < 0.05$) the seed germination rate of both weeds significantly in concentration dependent manner. Remarkably, under the treatment of high concentration (2.000 mg/mL), it can be observed that the germination of *L. perenne* and *B. pilosa* seeds were completely inhibited. Moreover, the germination index of seeds decreased with the increase of essential oils concentration, and the germination indexes of the two seeds were completely suppressed under the highest concentration (Figure 1C). Not only did it affect seed germination, but also had a certain effect on seedling growth (Figure 1D–F). Compared with blank control, except the seedling growth of *L. perenne* under the concentration of 0.125 mg/mL was greater than negative control, the growth of other seedlings gradually decreased under the increase of essential oil concentration. In particular, under the highest concentration of essential oil (2.000 mg/mL), *L. perenne* was completely inhibited (100%). However, it can be seen from the figure that the growth of seedlings was completely inhibited at 1.000 mg/mL and 2.000 mg/mL. It also can be seen that the effect of essential oil on the growth of *B. pilosa* seedlings was significantly greater

than that on the growth of *L. perenne* seedlings. In a word, essential oils have a certain dose effect on seed germination and seedling growth.

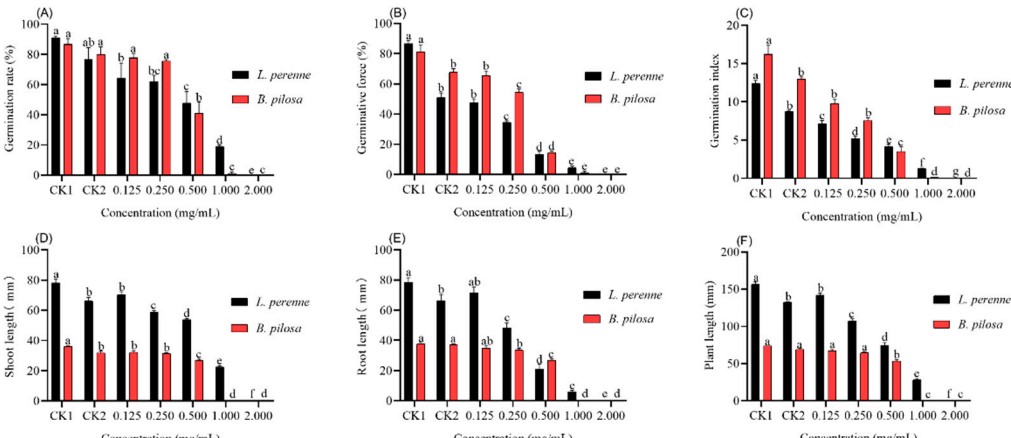

**Figure 1.** Effect of different concentration of *L. pungens* essential oil on seed germination and seedling growth (CK1, blank control; CK2, negative control; (**A**), Germination rate; (**B**), Germination force; (**C**), Germination index; (**D**), Shoot length; (**E**), Root length; (**F**), Plant length).

### 3.3. Effect of L. Pungens *Essential Oil on Photosynthetic Pigments Content (Petri Dish Experiment)*

Effects of different concentrations of essential oils on photosynthetic pigments of *L. perenne* and *B. pilosa* are shown in Figure 2A–C. The results show that the chlorophyll a and total chlorophyll content of the two plants decreased as the concentration of essential oil increased. Under the treatment of essential oils, the chlorophyll b content of the two plants did not change regularly, but the lowest chlorophyll content was still at the concentration of 1.000–2.000 mg/mL. It can be further seen from Figure 2 that the synthesis of photosynthetic pigments was affected after the treatment under high concentrations of essential oil (2.000 mg/mL), which in turn affected the photosynthesis of plants.

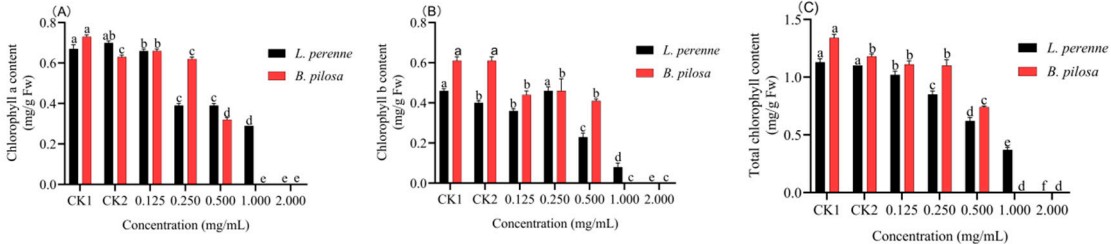

**Figure 2.** Effect of different concentration of *L. pungens* essential oil on photosynthetic pigments content (CK1, blank control; CK2, negative control; (**A**), Chlorophyll a content; (**B**), Chlorophyll b content; (**C**), Total chlorophyll content).

### 3.4. Pot Experiment

### 3.4.1. Seedling Growth Parameters

As shown in Figure 3, *L. perenne* caused obvious damage after spraying with different concentrations of *L. pungens* essential oil solution. On the fifth day of spraying, the leaves' etiolation and necrosis appeared. Especially, the effect on *L. perenne* was the greatest at 2.000 mg/mL. The results in Figure 4 show that the concentration of (0.250–2.000) mg/mL of *L. pungens* essential oil led to a significant ($p < 0.05$) decrease in seedling growth. Except at shoot length of 0.125 mg/mL and root length of negative control, other treatment groups showed a dose-dependent relationship with essential oil.

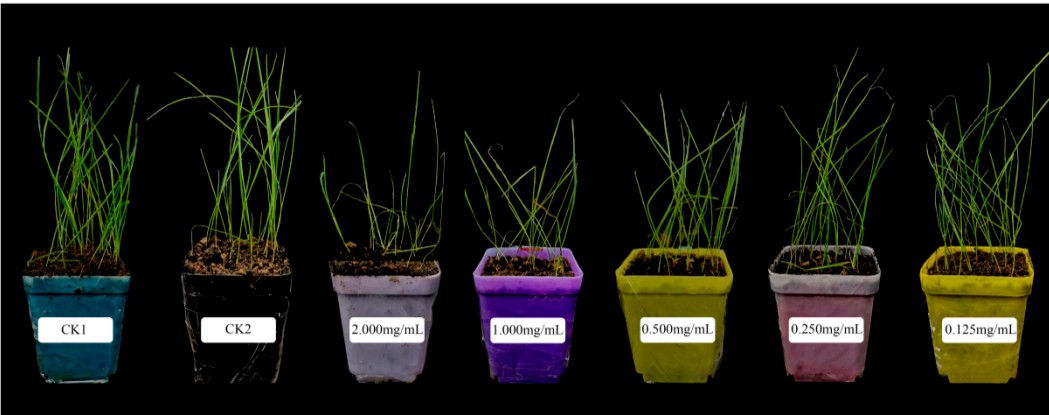

**Figure 3.** *L. perenne* in pot experiment (The concentration of potted plants from left to CK1 (blank control), CK2 (negative control), (2.000–0.125) mg/mL).

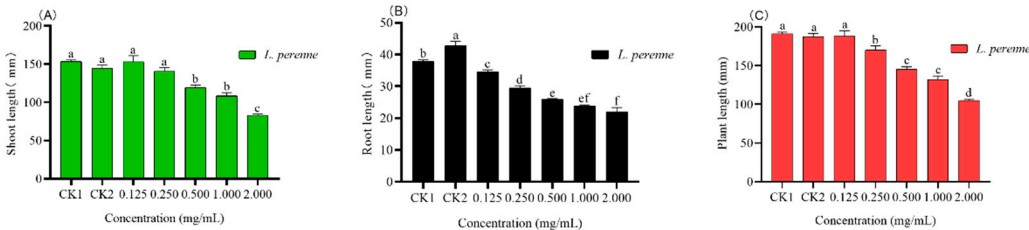

**Figure 4.** Effect of different concentration of *L. pungens* essential oil on seedling growth (CK1, blank control; CK2, negative control; (**A**), Shoot length; (**B**), Root length; (**C**), Plant length).

### 3.4.2. Photosynthetic Pigments Content

Chlorophyll and carotene are light-harvesting pigments, whose changes are related to photosynthesis. The results show that chlorophyll a, b, and carotenoid contents decreased with the increase of essential oil concentration (Figure 5A–D). In particular, under the concentration of 2.000 mg/mL, the difference was significant ($p < 0.05$) compared with the blank control, in which the content of chlorophyll a, b, total chlorophyll, and carotene decreased by 70.7%, 83.9%, 73.6%, 75%, respectively. Moreover, it can be seen from Figure 5 that except for the chlorophyll b measured after 0.125 mg/mL treatment, a similar decrease in all pigments suggests that the photosynthesis of *L. perenne* plants was inhibited, further affecting plant growth.

### 3.4.3. Malondialdehyde (MDA) Content and Electrolyte Leakage (EL)

The content of malondialdehyde (MDA) is the embodiment of the degree of membrane peroxidation in plant cells. After spraying with essential oils, the visible injury observed on *L. perenne* plants was presumed to be due to electrolyte leakage and membrane integrity. As shown in Figure 6A,B, different concentrations of essential oils had effects on MDA content and membrane leakage of *L. perenne*. The results show that the membrane leakage increased with the increase of MDA concentration, which was almost proportional to the concentration. Compared with the blank control, the difference is significant ($p < 0.05$), especially when the concentration of essential oil is 2.000 mg/mL, where MDA content and EL is the highest. Conclusively, the increase of the content of MDA and EL indicates that the degree of peroxidation of the plant cell membrane increases, accompanying by the gradually aggravating cell membrane damage.

### 3.4.4. Antioxidant Enzymes Activity

Catalase (CAT), superoxide dismutase (SOD), and peroxidase (POD) belong to antioxidant enzymes, and the levels of their activity are related to the growth of plants. The CAT, SOD and POD activity of *L. perenne*. were significantly affected ($p < 0.05$) in response

to spraying various concentrations of essential oils (Figure 7A–C). CAT enzyme activity increased first and then decreased, and the highest activity appeared at 0. 250 mg/mL (Figure 7A). The activity of the SOD enzyme was similar to that of the CAT enzyme, and the activity increased first and then decreased, among which the highest activity was at 0.250 mg/mL (Figure 7B). As shown in Figure 7C, the POD enzyme's activity increased with the increase of concentration, which was almost proportional to the concentration. It can be seen that the treatment of essential oil has a significant effect on antioxidant enzyme.

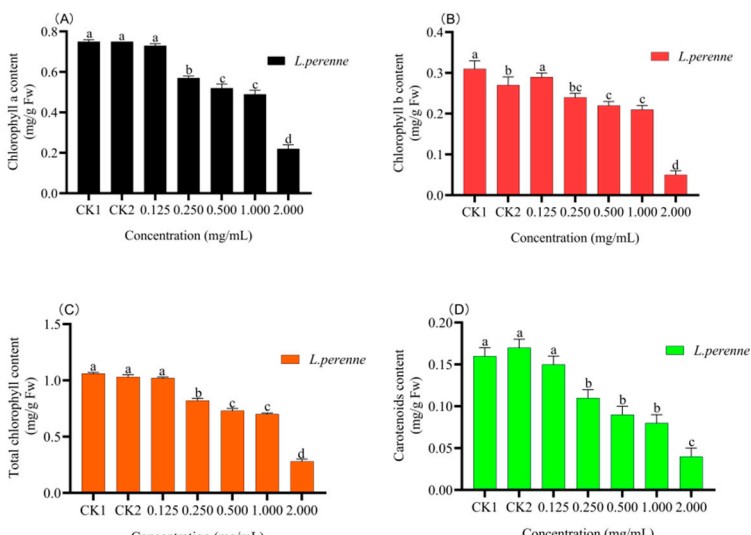

**Figure 5.** Effect of different concentration of *L. pungens* essential oil on photosynthetic pigments content (CK1, blank control; CK2, negative control; (**A**), Chlorophyll a content; (**B**), Chlorophyll b content; (**C**), Total chlorophyll content; (**D**), Carotenoids content).

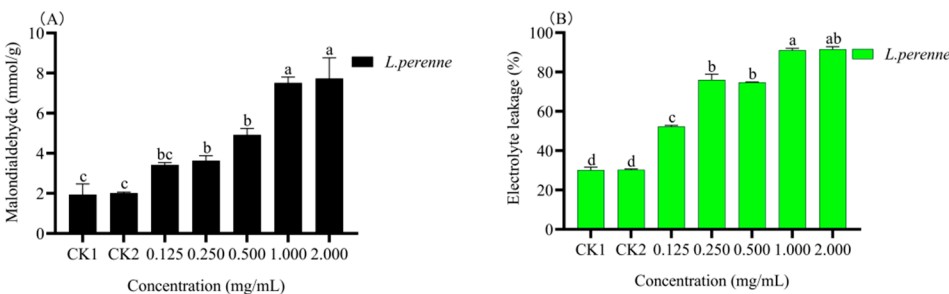

**Figure 6.** Effect of different concentration of *L. pungens* essential oil on malondialdehyde concentration and electrolyte leakage (CK1, blank control; CK2, negative control; (**A**), Malondialdehyde; (**B**), Electrolyte leakage).

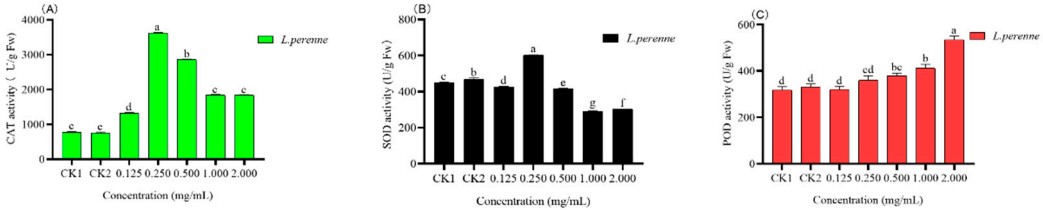

**Figure 7.** Effect of different concentration of *L. pungens* essential oil on catalase (CAT), superoxide dismutase (SOD) and peroxidase (POD) enzymes activity (CK1, blank control; CK2, negative control; (**A**), CAT activity; (**B**), SOD activity; (**C**), POD activity).

## 4. Discussion

In this study, the allelopathy of the *L. pungens* essential oil was evaluated. As far as we know, there are relatively few studies on the chemical components of the essential oil of *L. pungens*. In this study, limonene (30.29%), neral (29.44%) and linalool (6.38%) are the main components of the *L. pungens* essential oil. According to reports, Zhang et al. found that 1, 8-cineole was the main chemical component of the essential oil in the cotyledon of *L. pungens* [26]. Jiang et al. also studied and found that 1, 8-cineole was the most abundant essential oil in the cotyledon and twig of *L. pungens* [20], indicating that the essential oils extracted from different parts of one plant were different. Due to the lack of studies on *L. pungens*, we compared the components of the essential oil of *Litsea cubeba* which belongs to the same genus and has similar appearance and smell. Si et al. found that limonene (0.7–5.3%) in the essential oil of *Litsea cubeba* fruit exists only as a minor constituent in the essential oil [27], then Su et al. detected and analyzed the components of essential oil in the four parts of *Litsea cubeba*, flower, branch and fruit, and found that the main component of essential oil in leaf, flower and branch was 1, 8-cineole, and the main components of fruit were limonal and limonene [28]. It could be seen that there are similar differences in the main components of essential oil in different parts of *Litsea cubeba*. These differences may be related to the synthetic and metabolic differences of secondary metabolites in different parts of plant growth. The main components of the *L. pungens* essential oil are similar to that of *Litsea cubeba*, but the contents are different. It is speculated that the differences of these components may be caused by the genetic differences of species, or may be related to the differences of growth environment.

Essential oil from a natural plant is a valuable potential for allelopathic substance [6]. There are few research works on the bioactivities of the essential oil from *L. pungens*. Hence, the allelopathic activity in this work was firstly report. The results show that the essential oil had a strong allelopathy effect on both *L. perenne* and *B. pilosa*, especially at the concentration of 2.000 mg/mL, the germination and seedling growth of the two kinds of seeds were completely inhibited in the Petri dish test. In addition, the pot experiment also showed that after spraying *L. perenne* with different concentrations of essential oil, the plants appeared yellowing, necrosis and other phenomena, which were significantly different compared with the control group. According to literature reports, this phenomenon was also found in the previous study [29]. Angelini et al. reported that four monoterpenoids in three essential oils had a strong inhibitory effect on three annual weeds and three crop seeds [1]. Other monoterpenoids and sesquiterpenes in the essential oil from *Peumus boldus* and *Drimys winterii* were also found to exhibit an inhibitory effect on seed germination and seedling growth [30]. In addition, some researchers reported that terpenoids could change the permeability of cell membrane, enzyme activity, DNA transcription and RNA translation, thus inhibiting seed germination and seedling growth [31,32]. As shown in Table 1, terpenoids were the main components in the essential oil from *L. pungens*. Therefore, the allelopathy capacity of the essential oil may be associated with the high content of terpenoids.

After treatment with essential oil, the leaves of *L. perenne* were pale yellow and short, and the chlorophyll content decreased significantly, indicating that the essential oil treatment reduced the chlorophyll content of leaves and affected photosynthesis. These results are highly consistent with previous reports that some essential oils had a negative effect on chlorophyll content [10,33–36]. In addition, other works suggested that volatile allelochemicals might affect photosynthesis by reducing the formation of photosynthetic pigments [2]. However, further investigations are needed on whether the essential oil directly or indirectly interfered with photosynthesis.

In the life of plants, the balance of generation and elimination of reactive oxygen species (ROS) was regulated by the antioxidant enzyme system. Likewise, SOD and POD could scavenge the increasing ROS under acute stress [37]. Hence, the activities of antioxidant enzymes were closely related to the plant protection system [38–40]. In these work, the antioxidant enzyme activity of *L. perenne* was sharply reduced with the spraying

of essential oil from *L. pungens*, and the content of EL and MDA increased with the increase of the spraying time and concentration of essential oil. The accumulation of ROS could lead to membrane lipid peroxidation on the cell membrane, resulting in the loss of membrane function [41]. Moreover, the stimulation of monoterpenoids on these enzymes was found to lead to the overproduction of ROS [42]. In addition, monoterpenes also possessed strong lipophile properties, thus inducing membrane expansion, increasing fluidity, destroying membrane structure, and causing the inactivation of antioxidant enzyme system [35,43,44]. Therefore, the stimulation of monoterpenoids in the essential oil might account for the lower activity of antioxidant enzymes.

## 5. Conclusions

In conclusion, the essential oil of *L. pungens* has a strong allelopathic effect and a herbicide activity on weeds. We found that in the Petri dish experiment, the concentration of 2.000 mg/mL of essential oil completely inhibited the seed germination and seedling growth of *L. perenne* and *B. pilosa*. Secondly, in the pot experiment, it was also found that with the increase of the concentration of the essential oil of *L. pungens*, the chlorophyll content, EL, MDA, CAT, SOD and POD also had negative effects on the *L. perenne* seedlings. This experiment was the first to study the allelopathic effect of essential oil from *L. pungens*, which laid a foundation for the development of environment-friendly herbicides in the future and provided a new idea. Compared with commercial chemical herbicides, it is safer, non-toxic and easy to degrade. The applications of essential oils in these fields are very promising in the near future, so they are compatible with both environmental health and human health. However, further studies on allelopathic effects of single component or synergies between multiple components of *L. pungens* essential oil are needed to comprehensively evaluate potential herbicidal effects in order to determine its potential for large-scale use as a pesticide or herbicide.

**Author Contributions:** Formal analysis, S.F.; Investigation, Q.K., J.L., S.L. and X.W.; Methodology, T.X.; Project administration, M.Y. and T.C.; Software, H.X. and X.Z.; Supervision, C.D.; Writing—review & editing, L.Z. All authors have read and agreed to the published version of the manuscript.

**Funding:** This research received no external funding.

**Institutional Review Board Statement:** Not applicable.

**Informed Consent Statement:** Written informed consent has been obtained from all participants.

**Data Availability Statement:** The data that support the findings of this study are available from the corresponding author upon reasonable request.

**Conflicts of Interest:** The authors declare no conflict of interest.

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
