# Peer review of "Chemical Composition and Allelopathic Effect of Essential Oil of Litsea pungens"

_agronomy, doi:10.3390/agronomy11061115_

Round 1

Reviewer 1 Report

Report on Manuscript Number:

agronomy-1216292

Title:

Chemical composition and allelopathic effect of essential oil of Litsea pungens

For publication on:

Agronomy

Recommendation:

Minor Revision

Synthetic judgment

English style should be improved in some points.

queries

Answers to queries should be included in the paper where more appropriate.

Q1. Is the Petri dish the name of the experimental technique?

Q2. Row 314. Quality or quantity?

Comments, REMARKS, SUGGESTIONS

  • In reference to herbicides synthetic is a more correct adjective than chemical.
  • Table 1. Please, choose a criterion to list abbreviations of the classes of compounds (e.g. alphabetic or of first citation in the Table)
  • Row from (155 to 157). The sentence reported it is not completely true. In particular, at the higher concentrations the values are higher or lower than the control values.
  • Replace the title of paragraph 4.3 with Gas chromatography – Mass spectrometry ..
  • Please, use the name of the compound used not its trademark. Ethanol NOT Tween 80.
  • Check the Reference list. Some species names are not been written in italic.

General suggestions

Check accurately the whole paper, including tables, figure captions and references, to introduce some modifications listed below.

  1. Bidens pilosa NOT Pilosa
  2. Give the meaning of acronyms - both in the abstract and in the body of the article, independently - the first time they are cited. See CAT, SOD, POD,…
  3. Units and percentage symbol should be always written on the right side of the number, otherwise the use of parentheses is recommended. E.g. (0.250 to 2.000) mg/mL, (25 ± 2) °C.
  4. After a value the symbol of the unit should be reported, not the whole name. e.g. 6 h NOT 6 hours.
  5. Insert always a blank between number and unit or percentage symbol. 1 mL NOT 1mL.

Specific suggestions

  1. Row 58. Delete relative.

Row 236. Delete 

Reviewer 2 Report

Dear authors,

I had a great opportunity to review the research manuscript entitled “Chemical composition and allelopathic effect of essential oil of Litsea pungens” which is considered for publication in Agronomy Journal. I have analysed whole manuscript and it showed some interesting insights in topic of use of essential oil as bioherbicide. my opinion, the paper needs minor revision. Below I list several questions and comments about the manuscript that, in my view, will improve it. I recommend Authors to address them as best as they can.

Introduction

Please highlight briefly the problems caused by weeds in Agricultural field

"It is a new idea to replace chemical herbicides with allelopathy from plant sources, because it is safe, environmentally friendly, highly efficient, easy to degrade and difficult to develop drug resistance" Please rephrase this sentence (it not clear) and replace drug by herbicide. Natural products with allelopathic effects are not the only alternative to chemical herbicides, so please introduce other traditional methods that could also replace chemical products and then mention products from allelopathic plants.

Please explain the advantages and the disadvantages (e.g:  ecotoxicity at high concentrations) of the use of essential oils in agricultural systems.

Please introduce the different types of bioherbicide and place the essential oils in the appropriate category/type.

Materials and methods

Could you add the GPS coordinates of the collection site.

For Tween 80 (the final concentration was 1%), did you test its phytotoxicity?  Was it phytotoxic? If yes, please add more details to this point.

Line 297 : please add more details regarding the composition of the solution used as a controlhy you didn’t include used a positive control ( chemical herbicide or commercial bioherbicide as pelargonic acid). You can add this point in the conclusion ( further studies)

Please add the reason why you have used tween 80 in your solutions.

Results

To be more clear, please divide your trials into two sections : preemergence activity and post emergence activity. Adapt text and figures after modification.

Line 201 : which terpenoids ( monoterpene sesquiterpene) and how about their percentages

Table 1: It's very important to classify your identified compounds with percentage phenylpropanoids, monoterpene sesquiterpene).

Line 92-93 Please detail legend and describe all the treatments used in the trial. Idem for all figures.

Figure 3: Please describe in detail the photo, in particular the chlorosis and necrosis (e.g: after how much days the symptom appeared?, etc).

Discussion

For the postemergence test, Please detail the mode of action of the major compounds in the cuticle, membrane and wall of plants. It's mandatory to explain these effects since it explains how these compounds can cross the first barriers of leaves. 

I think it's mandatory to make a general scheme in which you detail at least one mode of action of your essential oils on plant growth. Please try to discuss all tested parameters referring to the scientific literature. You can use the following studies as references " Cynara cardunculus Crude Extract as a Powerful Natural Herbicide and Insight into the Mode of Action of Its Bioactive Molecules" and "Rosmarinus officinalis essential oil as an effective antifungal and herbicidal agent".

The oxidative stress is an indirect effect. Could you explain how the compounds identified in your EO could alter directly the biochemical processes of plants ? Please identify the multi-site action of your essential oil.

Please indicate at the end of the discussion the further studies or perspectives that you  propose to understand more the mode of action of essential oil in plants. Make a comparison with another commercial bioherbicide as pelargonic acid.

Good Luck 

Best regards

Reviewer 3 Report

A positive consequence of conducting this type of research will undoubtedly be, in the near future, the introduction of the obtained results to broad agricultural practice, especially to organic farms.
The work was done carefully, but the authors did not avoid a few errors that were noted in the manuscript.

Also....

The conclusions are written very generally. I am asking for a more detailed description of the research results obtained. Please provide more details. Just like it says in the abstract.
